# Remote near infrared identification of pathogens with multiplexed nanosensors

Robert Nißler [1,2], Oliver Bader [3], Maria Dohmen[1], Sebastian G. Walter[4], Christine Noll[3], Gabriele Selvaggio[1,2], Uwe Groß[3] & Sebastian Kruss [1,2,5 ✉]

Infectious diseases are worldwide a major cause of morbidity and mortality. Fast and specific detection of pathogens such as bacteria is needed to combat these diseases. Optimal methods would be non-invasive and without extensive sample-taking/processing. Here, we developed a set of near infrared (NIR) fluorescent nanosensors and used them for remote fingerprinting of clinically important bacteria. The nanosensors are based on single-walled carbon nanotubes (SWCNTs) that fluoresce in the NIR optical tissue transparency window, which offers ultra-low background and high tissue penetration. They are chemically tailored to detect released metabolites as well as specific virulence factors (lipopolysaccharides, siderophores, DNases, proteases) and integrated into functional hydrogel arrays with 9 different sensors. These hydrogels are exposed to clinical isolates of 6 important bacteria (*Staphylococcus aureus*, *Escherichia coli*,...) and remote (≥25 cm) NIR imaging allows to identify and distinguish bacteria. Sensors are also spectrally encoded (900 nm, 1000 nm, 1250 nm) to differentiate the two major pathogens *P. aeruginosa* as well as *S. aureus* and penetrate tissue (>5 mm). This type of multiplexing with NIR fluorescent nanosensors enables remote detection and differentiation of important pathogens and the potential for smart surfaces.

[1] Institute of Physical Chemistry, Göttingen University, Göttingen, Germany. [2] Physical Chemistry II, Bochum University, Bochum, Germany. [3] Institute of Medical Microbiology, University Medical Center Göttingen, Göttingen, Germany. [4] Department for Cardiothoracic Surgery and Intensive Care, University Hospital Cologne, Cologne, Germany. [5] Fraunhofer Institute for Microelectronic Circuits and Systems, Duisburg, Germany. ✉email: sebastian.kruss@rub.de

Microbial infections are one of the major causes of death in a global context. Often no or only limited diagnostic tools are available and treatment options are vanishing due to emerging antibiotic resistances[1,2]. One approach to counteract infections is their early detection and therefore there is a great need for fast and specific diagnostic tools. Additionally, tailored and personalized treatment pathways and antibiotic stewardship becomes increasingly important to reduce infection rates in hospitals and save lives and resources[3,4].

State-of-the art microbiological diagnosis[5] of bacteria relies on phenotyping characterization via cultivation on chromogenic media[6] in combination with DNA detection (PCR)[7] or mass spectrometry (MS) approaches[8]. Typical diagnosis times of these methods are on the order of several hours to several days[5]. Advancements in Raman spectroscopy and microfluidic lab-on-a-chip approaches aim to shorten time for diagnosis[9,10]. However, all these mentioned approaches require sampling, transport, purification, and/or cultivation. Therefore, not the analytical method itself limits time for diagnosis but rather multiple pre-analytical steps, which are necessary to receive, purify and process the sample. Label-free sensors could address this challenge by direct detection and identification of bacterial pathogens but need to be highly sensitive and selective to cover the diversity of potential pathogens and sample backgrounds[11–13].

Nanomaterials have been used to create highly sensitive biosensors[14,15]. For bacteria detection, different concepts including immobilization of antibodies against bacterial surface receptors and tailoring of electrostatic interactions have been employed[16]. However, remote optical detection with the desired selectivity and sensitivity remains a challenge. A versatile nanoscale building block for optical sensors are semiconducting single-walled carbon nanotubes (SWCNTs). They fluoresce without bleaching in the near infrared (NIR, 900–1700 nm) regime of the electromagnetic spectrum, thus offering tissue transparency due to decreased absorption and scattering, as well as ultra-low background fluorescence[17–19]. SWCNTs have been used as non-bleaching optical probes/sensors that are sensitive towards their chemical environment[20]. Such sensors were used to detect important small signaling molecules, nucleic acids, and proteins[21–24]. Furthermore, imaging many of them provides additional spatiotemporal information about biological

processes[25–27]. The key challenge in sensor development is tailoring their selectivity and sensitivity. SWCNTs have therefore been non-covalently functionalized with, e.g., proteins[28,29], peptides[30,31], single stranded (ss)DNA[30,32] or lipids[33] to achieve this goal.

The sensor requirements for the detection of bacteria are very high because infections/contaminations are a highly complex biochemical process and for example biofilm-mediated infections on implants are difficult to detect because samples are not directly accessible[34,35]. Additionally, one sensor alone could not be selective enough and the concept of a chemical nose appears to be more promising[36]. Therefore, fast and contact-free local detection without extensive sample taking and processing could advance the field of personalized pathogen diagnostics.

Here, we developed a set of NIR fluorescent nanosensors to remotely/directly identify and fingerprint clinically important bacteria.

## Results

**NIR fluorescent nanosensors for various bacterial motifs.** Bacteria are known to alter their chemical environment through the release of signaling molecules, enzymes, and metabolites[37]. Such molecules provide information about the nature of the bacterium. Especially virulence factors (e.g. exo- or endotoxins), signaling molecules (e.g. autoinducer or quorum sensing peptides), and matrix/biofilm materials can indicate the presence of specific bacteria[37–40]. However, a single molecular marker alone is unlikely to identify or at least narrow down bacterial species. Our approach is based on the idea that simultaneous detection of multiple analytes similar to an artificial nose increases sensitivity and selectivity of the analytical approach[36].

We therefore developed multiple NIR fluorescent nanosensors for different targets released by bacteria and incorporated them into biocompatible hydrogels (HG) onto which bacteria are plated (Fig. 1, step 1). Nine nanosensors were combined in a hydrogel array, which is remotely monitored by NIR stand-off detection (Fig. 1, step 2). This spatially encoded sensor pattern provides a NIR fingerprint of bacterial activity that is analyzed via multivariate data analysis (Fig. 1, step 3). In addition to spatial encoding, sensors could also be spectrally encoded (Fig. 1, step 4).

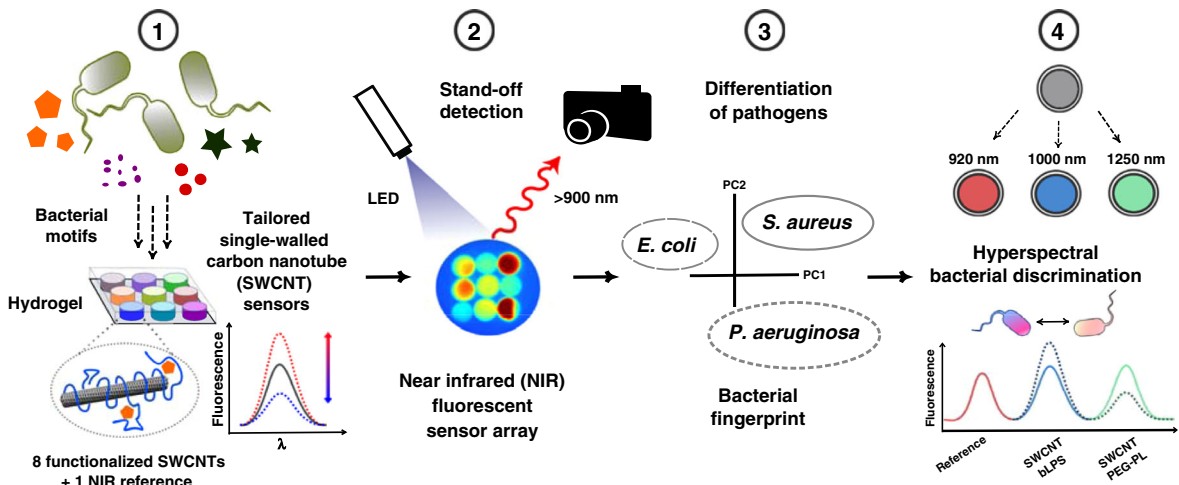

**Fig. 1 Remote detection of pathogens.** (1) Multiple nanosensors based on NIR fluorescent single-walled carbon nanotubes (SWCNTs) are synthesized in such a way that they change their fluorescence signal in response to bacterial metabolites and virulence factors (cell wall components, iron chelating molecules, secretory enzymatic activity). (2) Eight fluorescent nanosensors and one NIR fluorescent reference are incorporated into a polyethylene glycol hydrogel array that is remotely monitored in the NIR. (3) Bacteria growing on top of this hydrogel release molecules that change the (spatial) sensor array fingerprint, which allows us to differentiate important pathogens. (4) By using chirality-purified SWCNTs, multiple sensors can be spectrally encoded and used for hyperspectral differentiation of important bacteria such as *Staphylococcus aureus*.

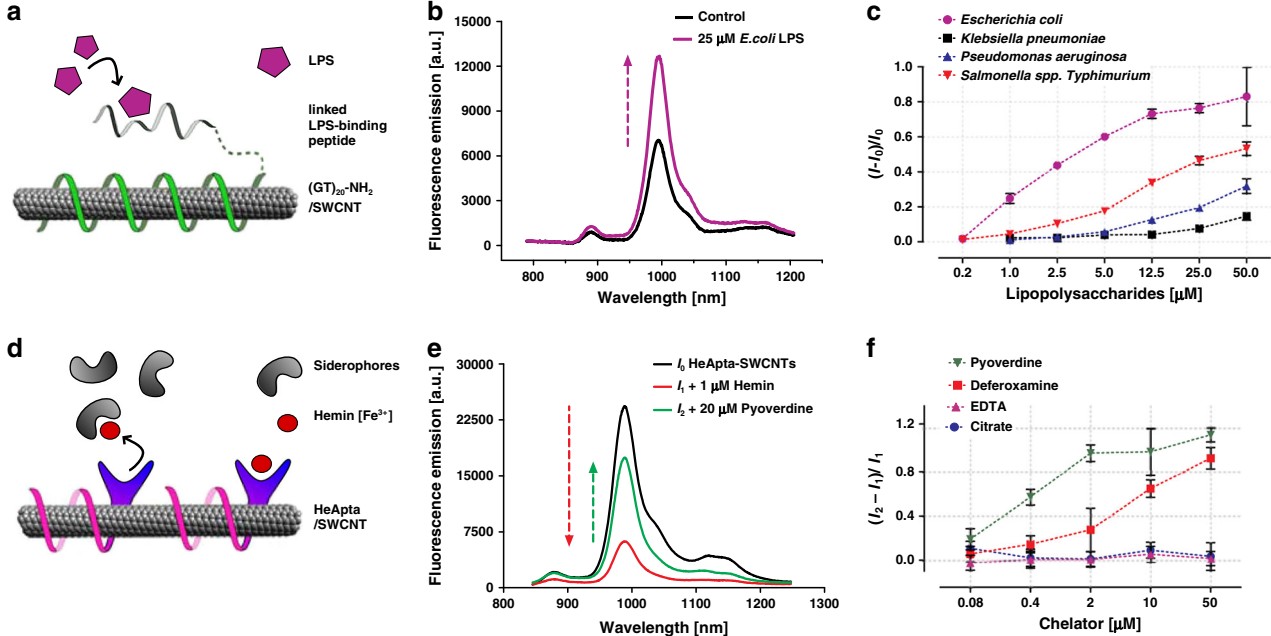

**Fig. 2 NIR fluorescent nanosensors of virulence factors. a** Design of an endotoxin sensor for bacterial lipopolysaccharides (LPS). $NH_2$-$(GT)_{20}$-ssDNA colloidally stabilizes the SWCNTs and was linked to a LPS-binding peptide via SMCC (succinimidyl 4-(N-maleimidomethyl) cyclohexane-1-carboxylate) chemistry. **b** NIR fluorescence increase of bindLPS-(bLPS)-SWCNTs after addition of 25 μM *E. coli* LPS. **c** Dose–response curve of bLPS sensors for LPS from *E. coli*, *K. pneumoniae*, *P. aeruginosa* and *Salmonella* spp. (n = 3 independent experiments, mean ± SD). **d** Design of the siderophore sensor. An aptamer (HeApta) binds hemin, which brings $Fe^{3+}$ into the proximity of the SWCNT and quenches it. Siderophores can reverse this effect by removing iron ($Fe^{3+}$), which increases fluorescence again. **e** Exemplary spectra of HeApta-SWCNTs. Addition of hemin ($I_0$ to $I_1$) quenches their fluorescence and addition of siderophores (pyoverdine) increases it again ($I_1$ to $I_2$). **f** Calibration of chelating agents with different stability constants ($K_f$) for iron ($Fe^{3+}$), added to HeApta-SWCNTs with 1 μM hemin concentration. Pyoverdine ($K_f = 10^{32}$), deferoxamine ($K_f = 10^{30}$), EDTA ($K_f = 10^{25}$), hemin ($K_f = 10^{22}$), citrate ($K_f = 10^{12}$)[78,89–91] (n = 3 independent experiments, mean ± SD).

The mentioned sensor array consists of eight SWCNT-based NIR fluorescent sensors of which four were tailored for specific bacterial targets and the other four are generic lower-selectivity sensors. Additionally, one very stable NIR fluorophore ($CaCuSi_4O_{10}$, Egyptian Blue-nanosheets, EB-NS) served as reference. For the specific sensors, we used rational design strategies to detect bacterial compounds and virulence-related enzymatic activity. The rational of using a mixture of specific and non-specific sensors was to reduce/account for background sensor responses in the final analysis. Additionally, the chemical complexity of the secreted substances makes it difficult to predict the overall performance and increasing the number of sensors appeared beneficial.

First, we developed a sensor that detects lipopolysaccharides (LPS), an endotoxin, which is part of the cell wall of Gram-negative bacteria and is shed into the bacterial environment[37,41]. For this purpose, a LPS-binding peptide[42–44] was conjugated to ssDNA/SWCNTs[30,45] (Fig. 2a, Supplementary Fig. S1a). The DNA adsorbs onto the SWCNT and translates conformational changes by LPS binding to the peptide into fluorescence changes. After optimization of the conjugation parameters (Supplementary Fig. S1), a colloidally stable conjugate could be created (bLPS-SWCNT). The NIR fluorescence of bLPS-SWCNTs increased (Fig. 2b) after the addition of *E. coli* LPS (76% for 25 μM). This fluorescence increase was concentration dependent and saturated at > 10 μM LPS (Fig. 2c) with a $K_d$ value of 1.87 μM (Supplementary Fig. S2). LPS from *Salmonella* spp., *P. aeruginosa* and *K. pneumoniae* showed similar but smaller fluorescence responses, indicating that the exact LPS structure[46] plays a role in fluorescence modulation. bLPS-SWCNTs also detect LPS when adsorbed onto a glass surface, which demonstrates that sensing is not based on aggregation or other colloidal effects in solution (Supplementary Fig. S2c).

Bacteria also release siderophores, which capture essential elements (e.g. iron or zinc) from their environment. These siderophores are important virulence factors and therefore targets for detecting bacterial pathogens[47,48]. Consequently, we designed a NIR sensor for siderophores (Fig. 2d) that is based on the idea that the removal of certain ions from the proximity of the SWCNT changes its fluorescence. Here, a hemin-binding ssDNA aptamer (HeApta) was adsorbed onto SWCNTs. Hemin addition quenched the SWCNT fluorescence, which can be attributed to the proximity of the iron ($Fe^{3+}$), complexed in the protoporphyrin IX (hemin), close to the SWCNT surface[24,49–51]. Stronger chelating agents such as the siderophore pyoverdine from *Pseudomonas fluorescens* (Fig. 2e, Supplementary Fig. S3) removed the iron and dequenched the NIR fluorescence. An optimal ratio of quenching by hemin and dequenching by pyoverdine addition was found at 1 μM hemin added to HeApta-SWCNTs ($A_{993nm} = 0.1$ for (6,5)-SWCNTs) (Supplementary Fig. S3b, c). This optimized siderophore sensor provides a concentration-dependent fluorescence increase for strong chelators ($K_f > 10^{30}$) such as pyoverdine ($K_d = 0.26$ μM) or deferoxamine ($K_d = 7.15$ μM) (Supplementary Fig. S3d). In contrast, weaker chelators such as ethylenediaminetetraacetic acid (ETDA) or citrate did not dequench hemin-HeApta-SWCNTs (Fig. 2f).

**Integration in hydrogel sensor arrays**. The rationally designed nanosensors for LPS and for siderophores are colloidally stable in solution. However, in complex media with many biomolecules, immobilized sensors should be even more resistant to unspecific effects such as aggregation or general degradation. This is especially relevant for sensors targeting enzymatic activity that rely on degradation of the organic functionalization around the SWCNT and would be prone to aggregation and precipitation in solution.

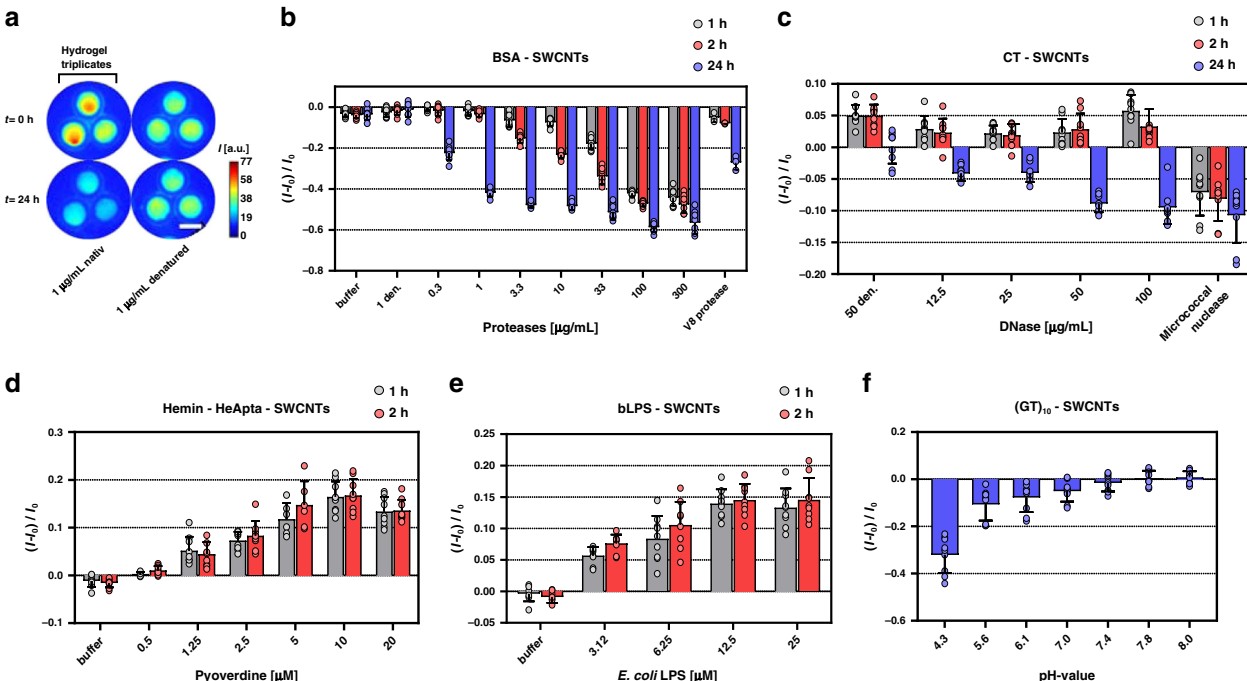

**Fig. 3 NIR fluorescent sensor hydrogels. a** NIR image of a polyethylene glycol hydrogel (PEG-HG) with embedded/copolymerized nanosensors in three identical regions (discs). Images were acquired remotely (distance 25 cm) with an InGaAs camera (see Fig. 4a for a picture of the setup). Here, only sensors reporting protease activity (see panel b) are depicted, but the concept applies to all sensors (scale bar = 0.5 cm). Note that the different NIR intensities of the discs are due to slight differences in illumination/imaging (distance/angle between sample and camera). **b** Protein (bovine serum albumin, BSA) functionalized SWCNTs, incorporated into a porous PEG-HG, decrease their fluorescence in response to protease from *Streptomyces griseus* (n = 3 independent experiments with three technical replicates each, mean ± SD) and V8 protease from *Staphylococcus aureus* (Endoproteinase Glu-C, 13.5 U/mL ~ 18 μg/mL) (n = 3 independent experiments, mean ± SD). **c** Long, genomic DNA molecules (denatured calf thymus (CT)-DNA on SWCNTs serve as substrate for nucleases. Incorporated into a porous HG, fluorescence decreases in response to native DNases I or *S. aureus* nucleases (11 UN/mL ~ 55 μg/mL) (n = 3 independent experiments with three technical replicates each, mean ± SD). **d, e** Tailored nanosensors (see Fig. 2) are still functional when incorporated into a hydrogel (n = 3 independent experiments with three technical replicates each, mean ± SD). **f** (GT)$_{10}$-SWCNTs (as one of the generic DNA/SWCNT sensors) in a HG shows a pH-dependent fluorescence response (evaluated after 24 h) (n = 3 independent experiments with three technical replicates each, mean ± SD).

Therefore, we incorporated these sensors into porous HGs based on biocompatible poly(ethylene glycol)diacrylate hydrogels (PEG-HGs). HGs of low (type-I) and high porosity (type-II) (Supplementary Fig. S4, Table ST1, ST2) were created by using PEG-DA (700 g/mol), in combination with different concentrations of higher molecular weight PEG[52]. The rationale was that (type-II) gels would allow large enzymes to diffuse into the gel and reach the nanosensors. In contrast, for small molecules such as siderophores type-I gels are used to let relevant analytes pass through but prevent at the same time unspecific effects. As a first target, extracellular proteases were chosen[53–55]. For this purpose, SWCNTs were modified with bovine serum albumin (BSA), serving as an enzymatic substrate, and incorporated into porous (type-II) HGs. When the sensor gels were incubated with a serine protease from *Streptomyces griseus* (Fig. 3a), fluorescence decreased by more than 40% within 24 h in the presence of 1 μg/mL native protease compared to the negative control (thermally denatured protease). Additionally, fluorescence spectroscopy of the HGs revealed that the emission of (6,5)-SWCNTs shifted by 5–7 nm into the red (Supplementary Fig. S4) suggesting decomposition of the BSA surface coating. The fluorescence signal decreased faster for higher protease concentrations, resulting in an EC$_{50}$ = 0.4 μg/mL for 24 h (Fig. 3b, Supplementary Fig S5). Another relevant protease from *S. aureus* (V8) showed the same response (Fig. 3b). Following the same principle, a sensor for nuclease activity was designed, which is an important virulence factors of *S. aureus*[56]. Micrococcal nuclease from

*S. aureus* is known to degrade single-stranded calf thymus (CT) DNA[57] and therefore we used CT-ssDNA to functionalize and disperse SWCNTs (Supplementary Fig. S6). CT-SWCNTs were then incorporated in (type-II) HG and were able to report both DNase I and *S. aureus* nuclease activity (Fig. 3c). DNase I addition (12.5–50 μg/mL) increased fluorescence on short time scales (1 h) but furthermore reduced fluorescence for longer time scales (−10% for 100 μg/mL after 24 h). Addition of thermally denatured DNase I (50 μg/mL) did not decrease fluorescence, indicating that only active enzymes affect CT-SWCNTs fluorescence significantly. Micrococcal nuclease on the other hand directly decreased the fluorescence of CT-SWCNTs within 1 h. Therefore, it seems likely that target site specificity of different nucleases will cause different sensor responses and kinetics[58], a potential basis for the development of more specific sensors in the future.

All sensors including the colloidally stable ones (Fig. 2) were integrated into HGs to create a functional sensor material for NIR stand-off detection (HG sensor spot diameter = 5 mm, HG array: 15×15×0.8 mm). Sensors that did not require immobilization into a HG in the first place such as HeApta-SWCNTs sensors were integrated into type-I-HGs to exclude unspecific protein adsorption effects, but allow smaller molecules such as siderophores to reach the SWCNTs. This procedure had to be optimized to obtain highly fluorescent HGs (Supplementary Fig. S7). Similar to the solution experiments, HeApta-SWCNT HGs increased in response to pyoverdine (~1200 Da) (Fig. 3d), saturating at ~10 μM.

No further fluorescence change was observed for timepoints >1 h (Supplementary Fig. S8a, b) indicating a diffusion limited response within the first few minutes. Similarly, bLPS-SWCNTs were integrated in macroporous type-II-HG, to enhance diffusion of the ~10 kDa large target analyte LPS[59]. HG fluorescence increased upon E. coli LPS addition and saturated at a concentration of 12.5 μM within 20–40 min (Supplementary Fig. S8c).

The four sensors described above were each developed in a rational way to target specific bacterial moieties. Furthermore, pH changes due to metabolic activity of bacteria could be another marker and ssDNA-SWCNTs are known to respond to the proton concentration[60]. Consequently, we incorporated $(GT)_{10}$-SWCNTs into type-I-HG and the NIR fluorescence of the resulting sensor HGs decreased with pH (Fig. 3f, Supplementary Fig. S8d) by more than 30% at pH 4.3. Such sensor HG reports therefore pH changes or could serve as reference for other sensors that are affected by metabolic acidification.

It is known that small changes in the chemical functionalization (e.g. DNA sequence) of SWCNTs change their selectivity to different small molecules[61]. Therefore, three other sensor hydrogels based on $(C)_{30}$- and $(GC)_{15}$-ssDNA as well as PEG-phospholipid (PEG-PL)-functionalized SWCNTs were created to further increase the multiplexing level. These sensors did not target specific analytes but are known to react to potential changes in pH[60], oxygen concentration[62] or to increasing protein concentrations[23] (see Supplementary Table ST3). Therefore, we hypothesized that characteristic fluorescence changes even if not directly related to one target molecule could increase the discrimination power of the sensor array and decrease the impact of background signals. Last, we added a reference hydrogel with incorporated nanosheets of the calcium copper silicate Egyptian Blue (EB-NS, $CaCuSi_4O_{10}$) as a highly stable reference NIR fluorophore at the lower end of the NIR emission capabilities of SWCNTs (emission at $\lambda \approx 920$ nm)[63]. All 9 sensor hydrogels were then assembled into a stable $3 \times 3$ HG array, suitable for further integration into microbiological agar (Supplementary Figs. S9–S11 and Table ST3). To avoid contamination, the hydrogels were disinfected by UV light before experiments.

**Remote NIR imaging of bacteria.** These hydrogels (embedded in agar) were used as local sensors for bacteria and were remotely imaged (stand-off detection) in a simple optical setup that is portable and could be transported into the labs with higher biosafety that were necessary to work with pathogens from patients. It consists of a NIR sensitive InGaAs camera, a LED white-light source with 700 nm short pass filter, an objective lens and optical filters for NIR light (>900 nm). To test the sensors ability to distinguish different bacterial species, each sensor array (Fig. 4b, c) was challenged with bacterial suspensions (in the same medium) to mimic exposure to bacteria, metabolic activity and biofilm formation (100 μL 0.5 McFarland standards) of six different pathogens (S. aureus, S. epidermidis, S. pyogenes, E. faecalis, E. coli, and P. aeruginosa). These pathogens (reference strains and clinical isolates from patients) are amongst the most prominent bacteria causing post-surgery infections in artificial joint implants, for which remote optical detection could be a promising tool[64,65]. During bacterial metabolic activity and growth, the NIR fluorescence of the sensor array was imaged remotely (25 cm) in a direct and non-destructive way. Exemplary NIR images during incubation with S. aureus indicated significant fluorescence changes over time (Fig. 4d). The corresponding sensor responses ($\Delta I_{SR}$) were normalized to the EB-NS reference fluorophore, and differences increased over time as expected (Fig. 4e). These sensor patterns served as fingerprints (see data for all tested bacteria in Supplementary Fig. S12) and showed prominent differences

between different pathogens (final 72 h timepoint: Fig. 4f, g). Generally, the presence of bacteria altered the pattern of the sensor array towards either increased (S. aureus, S. epidermidis, or P. aeruginosa) or decreased fluorescence (E. faecalis or S. pyogenes). Next to the interspecies differences, isolates from the same bacteria species varied in response e.g. P. aeruginosa and E. coli (Supplementary Fig. S12, S13). However, the presentation of the data in Fig. 4g is not optimal to highlight and distinguish different bacterial species. Consequently, a multivariate statistical analysis (principal component analysis, PCA) was performed (Fig. 4h), which revealed a time-dependent separation of clusters (0.68 bivariate ellipse confidence interval) corresponding to different bacterial species. Bacterial growth and metabolic activity did not strongly change the sensor array response within the first 12 h, possibly limited by release and diffusion of the target molecules into the sensor gel. Within 24 h P. aeruginosa and S. aureus / S. epidermidis clusters separated from the control. After 36 h, additionally P. aeruginosa, S. aureus, S. epidermidis, and E. coli clusters separated. Only E. faecalis and S. pyogenes could not be distinguished, even after 72 h. For different strains from one pathogen sub-clustering was observed, highlighting that the sensor array can not only distinguish pathogens species, but possibly even different strains from various clinical sources of the same species (Supplementary Fig. 13).

To test the medical relevance and potential, clinical isolates (n > 20) from S. aureus and S. epidermidis were analyzed. Both species are responsible for over 50% of all clinical joint infections[64]. The isolates were chosen to get a broad distribution of differences based on genotyping to cover a diverse population (Supplementary Table ST5). The sensor response from all isolates after 72 h incubation is shown in Fig. 4i. Both bacterial species caused similar response patterns that differed in mean intensity, with a certain variation in between the isolates (Supplement Fig. S14). PCA revealed that the two different bacteria populations can be distinguished (Fig. 4j). Both populations separated into two clusters with a small overlap, indicating that the majority of the tested isolates yield a similar sensor response and only a few isolates skewed the separation (extended dataset in Supplementary Fig. S15). Furthermore, when using the spectral fingerprints from all 43 clinical isolates as a trainings dataset for linear discriminant analysis (LDA), the analyzed S. aureus and S. epidermidis fingerprints from Fig. 4g could be classified and assigned with a ~80% likelihood (Supplementary Fig. S15c). As seen from these experiments, the magnitude of the sensor response depends on incubation time. However, this is not the real time-resolution of the sensor but rather reflects diffusion in the HG and metabolic rates of the different isolates. To evaluate the timescale on which the sensor array responds (see also Fig. S8), bacterial culture supernatants were added to the sensor array and monitored. For P. aeruginosa (Fig. 4k) and S. aureus (Supplementary Fig. S16) the sensor array responded between 15 and 45 min after addition with a specific pattern. The results shown in Fig. 4 raise the question if additional sensors could further increase the analytical performance of the sensor gel. To get a quantitative estimate we developed a stochastic simulation that predicts how the discrimination power scales with the number of sensors (Fig. 4l). It is based on the assumption that one can develop additional sensors in the experimental range found by us including non-responsive sensors, noise and typical sensor responses (see "Materials and Methods" for details and Supplementary Fig. S17). The results indicate that the analytical performance of the 9-sensor array could be further increased by more sensors but the gain would decrease for 15 sensors or more. For point-of care diagnostics, the overall size of an array and the number of sensors are competing features and this simulation provides a quantitative way for optimization.

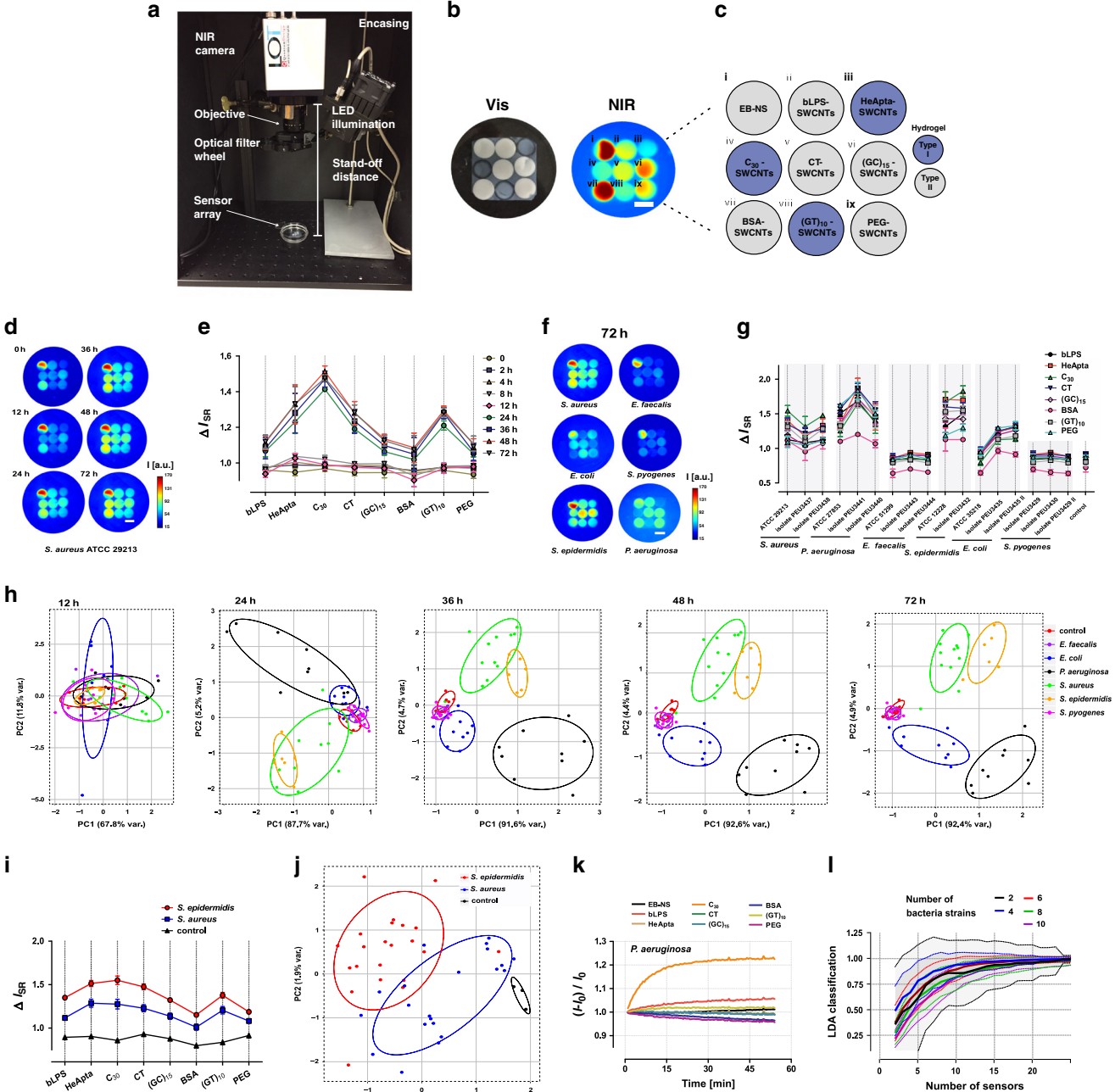

**Fig. 4 Remote NIR identification of bacteria. a** A simple NIR stand-off setup enables remote (25 cm) imaging of the NIR fluorescent sensors embedded in a hydrogel (HG) array. **b** Photograph (in the visible spectrum) of the HG nanosensor array and its corresponding NIR fluorescence image (scale bar 0.5 cm). **c** Arrangement and functionality of the 9 sensors in the HG array. **d** Remote NIR fluorescence image of a sensor array incorporated in a microbiological agar plate, inoculated with *S. aureus*. During bacterial growth the sensor pattern changes (scale bar 0.5 cm). **e** Corresponding sensor response normalized to the EB-NS signal during *S. aureus* growth, from 0 to 72 h ($n = 3$ independent experiments, mean ± SD). **f** Representative fluorescence response fingerprint of six pathogens, monitored after 72 h. **g** Fluorescence SWCNT array fingerprint of all bacteria and strains, evaluated after 72 h. ($\Delta I_{SR}$—sensor response: $I_{S1}/I_{R1}/I_{S0}/I_{R0}$; $I_S$—intensity sensor, $I_R$—intensity reference (EB-NS)) ($n = 3$ independent experiments, mean ± SD). **h** PCA (principal component analysis) of the fluorescence fingerprint of all analyzed strains, plotted for different timepoints (12–72 h). Each point represents one bacterial sample including clinical isolates from different patients. Control = medium only. **i** Mean sensor array fingerprint from diverse clinical isolates of each *S. aureus* ($n = 21$ biologically independent samples) and *S. epidermidis* ($n = 22$ biologically independent samples) 72 h after incubation. (error = SE). **j** Corresponding PCA for the array fingerprint after 72 h growth of the clinical isolates of *S. aureus* and *S. epidermidis*. **k** Time resolved fluorescence change of the nanosensor array after addition of liquid culture supernatant from *P. aeruginosa*. (24 h incubation in LB-medium, *I*—intensity sensor at $t = x$; $I_0$—intensity sensor at $t = 0$) (mean of $n = 3$ independent experiments). **l** Stochastic simulation that predicts how bacteria discrimination improves with number of sensors. The simulation is based on experimental responses and selectivities as range for novel sensors and uses PCA as well as mean linear discrimination analysis (LDA) to distinguish bacteria. Mean values are plotted and the dashed lines/transparent area represent the SD from 25 independent simulations. Ellipses in (**h**, **j**) indicate the 0.68 bivariate confidence interval.

Overall, this multiplexing sensor array was able to detect the presence of bacteria and differentiate a majority on the species level, based on their metabolic fingerprint. Even closely related important pathogens isolated from diverse human infections (*S. aureus* and *S. epidermidis)* could be distinguished.

To evaluate the sensor array performance in the context of smart surface applications such as in implants, host-induced background responses were tested using human synovial fluid (Supplementary Fig. S18). The overall sensor response was not affected when synovia from in total 26 healthy and infected patients were compared, which indicates no interfering immune response background that could bias fingerprinting (Supplementary Fig. S18, Table ST4). Furthermore, bacterial targets like proteases or metabolism induced pH changes could be sensed in the presence of the synovia, while even sensing of methicillin-resistant *S. aureus* (MRSA) was possible in the synovial milieu (Supplementary Fig. S19). We concluded that the sensor array could respond towards a local, biofilm-based infection, while background signals in synovia would not lead to a false-positive readout.

**Hyperspectral NIR detection of bacteria.** In the array presented above, the different sensors are spatially encoded, which is useful for point-of-care in vitro bacteria diagnostics. However, for smart materials or in vivo applications spectrally encoded sensors would be beneficial. They would enable ratiometric detection and hence decrease problems due to inhomogeneous illumination, spatial resolution, etc. To achieve spectral multiplexing, SWCNTs are needed that do not overlap in their fluorescence emission (i.e. different SWCNT chiralities). Even though a lot of progress was made in the last decade in SWCNT purification[66–69], it is still an ongoing area of research and sensing with purified SWCNT has only been shown in a few cases[70,71].

To evaluate spectral multiplexing, three different sensors from the 9-sensor array were used to distinguish *S. aureus* and *P. aeruginosa*. Indeed, bacterial differentiation was still possible even with a reduced number of sensors (Supplementary Fig. S20 and Fig. S21) that differed most for different pathogens. bLPS- and PEG-SWCNTs showed distinct responses for *S. aureus* and *P. aeruginosa* (Fig. 4g) and were therefore chosen for spectral multiplexing. EB-NS (~920 nm emission) served again as NIR reference fluorophore. SWCNT chiralities were separated by aqueous two-phase extraction (ATPE), yielding monodisperse CoMoCAT (6,5)-SWCNTs (980 nm emission) and larger-diameter HiPco-SWCNTs chiralities (emission > 1110 nm) (detailed information in Supplementary Fig. S22 and S23). By surface exchange of the purified nanotubes, bLPS-(6,5)-SWCNTs and PEG(5 kDa)-PL-(9,4),(8,6),(9,5)-SWCNTs (Supplementary Fig. S22 and S23) could be created. These two different SWCNT sensors and EB-NS were incorporated together into a HG. Consequently, each sensor could be read out at a different wavelength by switching the emission filter (Fig. 5a) in the stand-off setup (Supplementary Fig. S24). This functional sensor HG (Supplementary Fig. S25) was integrated in microbiological agar and inoculated with *S. aureus* and *P. aeruginosa* (one reference strain and two clinical isolates), as described before. Clear differences were observed between the two species and also for isolates of *P. aeruginosa* (Fig. 5b). Similar to the spatially encoded sensor arrays, PCA revealed clusters that were fully separated after 72 h (Fig. 5c). The results indicate that the major spread within one bacterial cluster is due to the biological difference between the tested strains.

For a future smart implant application and in situ diagnostics, one major advantage of the NIR is tissue penetration. Consequently, we tested how deep we can probe such sensors especially because this would be a requirement for medical applications (e.g. sensors in artificial (knee) joint implants or venous catheters). Fluorescence decreased with thickness of a tissue phantom (chicken) (Fig. 5d, Supplementary Fig. S25) but at moderate excitation intensities (25 s, 0.176 W mm$^{-2}$) signals from below 7 mm thick tissue were detected (Fig. 5e). By using higher excitation energies and advanced imaging approaches such as pulsed laser illumination or fluorescence lifetime imaging, this level of tissue penetration could be further increased and enable in vivo applications especially in tissue close to the body surface. For deep-tissue applications in humans one could also make use of light-guides or miniaturized endoscopes. Additionally, due to the structure-dependent fluorescence emission wavelength of SWCNTs, one could envision up to around 15 spectrally different SWCNT sensors in the NIR range[72].

**Discussion**

Bacterial infections require timely treatment and local/fast detection is one of the great challenges in biomedicine. Here, we developed multiple NIR fluorescent sensors to remotely finger-print important pathogens. The SWCNT-based sensors were engineered to detect bacteria via their secreted metabolites. This approach is different from concepts that detect genetic information (PCR) or the chemical composition of the bacteria itself (MS, Raman spectroscopy). The nanosensors detect major bacterial virulence factors (LPS, siderophores), as well as enzyme activity (DNases and proteases) and generic metabolic activity and are embedded in hydrogels that are remotely imaged in the NIR. The SWCNT's NIR fluorescence makes these nanosensors an ideal tool for non-invasive, fast and local identification of bacterial infections and contaminations. Spatial encoding of nine different sensors allowed to fingerprint pathogens such as *E. coli*, *S. aureus* or *P. aeruginosa* after 24–72 h on the species level. The finger-prints of 43 additional clinical isolates of *S. aureus* and *S. epidermidis* showed that even closely related bacterial species could be distinguished. The analysis of the sensor array pattern could be further improved by using more sophisticated machine learning algorithms[36,73]. Especially if the number of sensors is further increased such concepts would further improve and accelerate precise classification and identification of bacterial contaminations. In contrast to previous approaches, the developed sensors detect secreted bacterial motifs and are not only labels[74]. Multi-plexing with non-SWCNT nanosensors has been used before to distinguish non-pathogenic from pathogenic biofilms[75]. However, the advantages of sensors that fluoresce in the NIR enable effective remote imaging in relevant distances (25 cm) or under tissue without the typical background fluorescence found in the visible of the electromagnetic spectrum. Additionally, a major advantage of these sensors is that their sensitivity/selectivity can be easily modified by changing the surface chemistry e.g. by using different DNA sequences. Consequently, upscaling the number of sensors is only limited by practical aspects such as the lateral size of the sensor array. The standoff imaging of the bacteria sensors presented in this work is not limited to smart surfaces in point-of-care tools, hospitals or implants but could be expanded to detect also bacterial infections (in plants) that reduce yields in agriculture[24,74,76].

The modular chemical design of the SWCNT functionalization is useful to create more sensors and increase the multiplexing level and thus sensor performance. In this context, the advent of covalent functionalization of SWCNTs with biomolecules without impairment of NIR fluorescence will open up additional possibilities[77]. For point-of-care diagnostics the current time resolution should be further increased. It is mainly limited by the diffusion of the analytes through the agar layer and the sensor

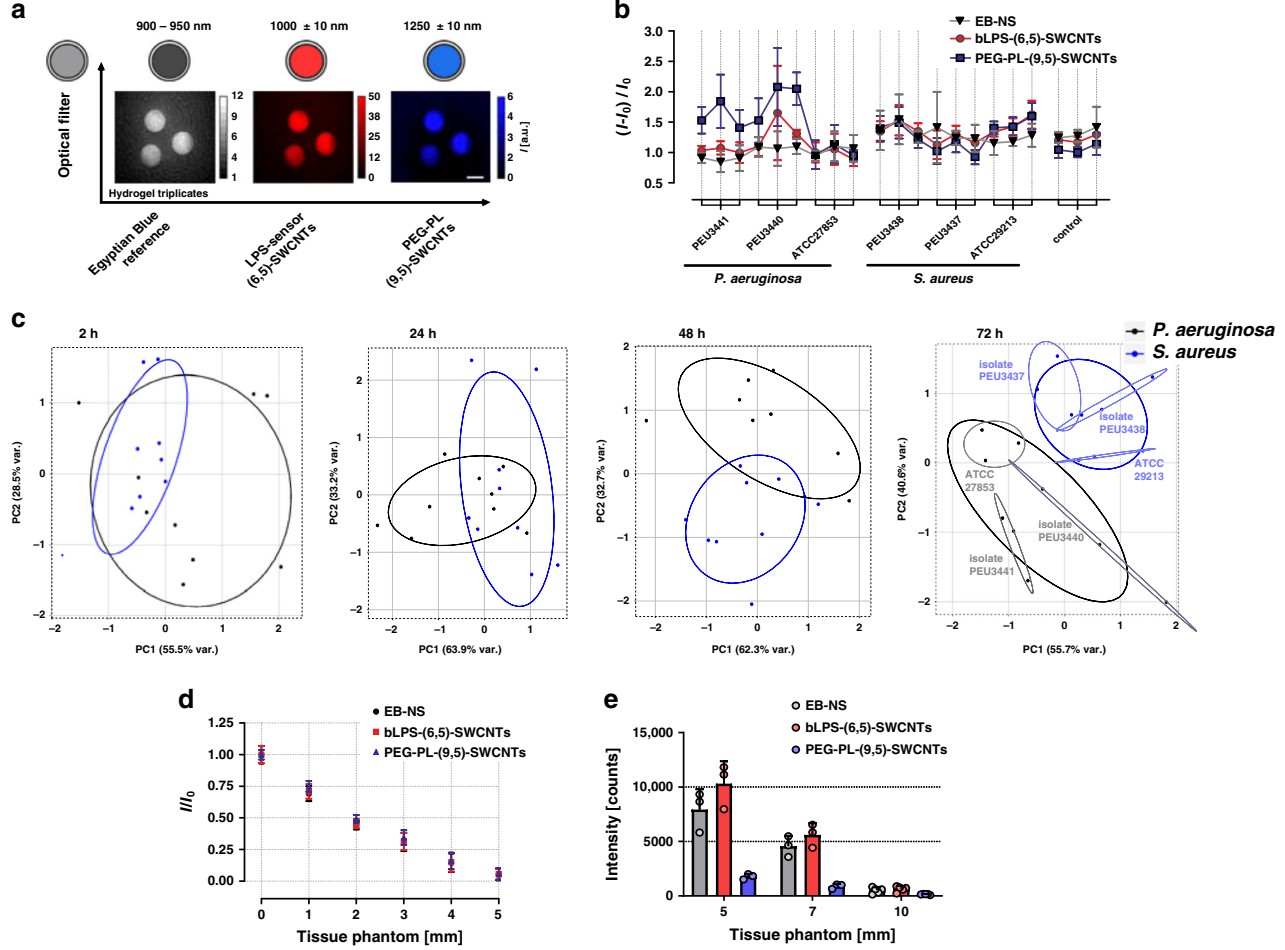

**Fig. 5 Hyperspectral remote detection of bacteria. a** Multi-color sensor HGs are created by incorporating EB-NS as reference NIR-fluorophore and two chirality-purified SWCNT sensors (LPS-binding-peptide-(GT)$_{20}$@(6,5)-SWCNTs and PEG(5 kDa)-PL@(9,5)-SWCNTS), which enables spectral multiplexing. NIR fluorescence images were captured with optical filters, resulting in three emission/color channels (EB-NS 900–950 nm, bLPS 1000 nm and PEG 1250 nm). Note that heterogeneity in fluorescence of the three technical replicates is caused by inhomogeneous illumination intensity and not relevant for quantification because of normalization to the EB-NS reference (scale bar 0.5 cm). **b** Fluorescence change of the hyperspectral sensors after 72 h of incubation with *P. aeruginosa* and *S. aureus* show distinct responses for one reference strain and two clinical isolates (n = 3 independent experiments, mean ± SD). **c** PCA of the spectrally encoded sensors for different timepoints. Each point represents one biological replicate of the indicated strain. Ellipses indicate the 0.68 bivariate confidence interval. **d** Tissue penetration through chicken phantom (561 nm excitation, 130 mW, 6 s integration time, *n* = 3 sensor spots, mean ± SD). Intensity decreases with tissue thickness. **e** If the integration time is increased (25 s, 190 mW excitation power) spectrally encoded bacterial sensors can be read out through 7 mm of tissue phantom (*n* = 3 independent experiments, mean ± SD).

hydrogel. Gel thickness as well as lateral sizes of sensor spots can be further miniaturized to increase time resolution and sensitivity. Such advances could facilitate fast in vitro testing without the need for large laboratory equipment and enable e.g. blood-culture based sepsis diagnostics. In contrast to the array, hyperspectral imaging will be limited to a smaller number of sensors. However, ratiometric imaging and detection as shown for the two major pathogens *S. aureus* and *P. aeruginosa* promises remote detection and is required for potential in vivo applications such as smart implants that would especially profit from NIR light. In the long-term, these developments could facilitate in situ diagnostics of infections in non-accessible locations such as on implants.

In summary, we developed NIR fluorescent nanosensors to remotely fingerprint bacteria. The combination of multiple sensors with different selectivities allowed us to distinguish clinically relevant bacteria based on their metabolic fingerprint. Multiplexing was achieved by spatial or spectral encoding, which highlights the opportunities for remote pathogen detection. In the future, NIR remote detection of bacteria could enable faster diagnostics and tailored antibiotic treatment, which would

ultimately result in better clinical outcomes and lower mortality rates.

## Methods

**Materials**. All materials, if not otherwise stated, were purchased from Sigma Aldrich.

**SWCNT surface modification**. (6,5)-chirality enriched CoMoCAT SWCNTs (Sigma Aldrich) were used and modified with different macromolecules. Functionalization with ssDNA such as (GT)$_{10}$, $C_{30}$ and (GC)$_{15}$ (oligonucleotide sequences purchased by Sigma Aldrich) followed a previously described protocol[60]. In short, 100 µL ssDNA (2 mg/mL in PBS) were mixed with 100 µL PBS and 100 µL SWCNTs (2 mg/mL in PBS), tip sonicated for 15 min @ 30% amplitude (36 W output power, Fisher Scientific model 120 Sonic Dismembrator) and centrifuged 2 × 30 min @ 16,100 × *g*. If stated, the excess ssDNA was removed by molecular weight cut-off filtration (100 kDa, Sartorius). Supplementary Table 2 provides an overview of the conditions used for SWCNT modification. For the HeApt-SWCNTs the hemin-binding aptamer 5′- AGT GTG AAA TAT CTA AAC TAA ATG TGG AGG GTG GGA CGG GAA GAA GTT TAT TTT TCA CAC T-3′ was used.[49,50] To modify the SWCNT surface with a long, genomic ssDNA, 3 mg/mL calf-thymus (CT) DNA was beforehand denatured for 30 min @ 100 °C. Phospholipid-PEG-SWCNTs were synthesized by performing dialysis of sodium cholate suspended SWCNTs.[33] For this purpose, SWCNTs were tip-sonicated in

500 µL (10 mg/mL in PBS) sodium cholate. After centrifugation, 200 µL supernatant was mixed with 800 µL sodium cholate (10 mg/mL) containing 2 mg 18:0 PEG5000 PE (1,2-distearoyl-sn-glycero-3-phosphoethanolamine-N-[methoxy(polyethylene glycol)-5000], Avanti Lipids). The mixture was transferred to a 1 kDa dialysis bag (Spectra/Por®, Spectrum Laboratories Inc.) and dialyzed for several days against 1xPBS. Centrifugation 30 min @ 16100x g yielded the colloidal dispersed PEG-PL-SWCNTs.

Exfoliated Egyptian Blue nanosheets (EB-NS) were obtained by following a modified version of the previously reported protocol from Selvaggio et al.[63]. EB powder (Kremer Pigmente GmbH & Co. KG.) was milled by means of a planetary ball mill (PB, Pulverisette 7 Premium Line, Fritsch, Germany) equipped with 20 mL agate beakers and 5 mm agate balls, in deionized $H_2O$ at 900 r.p.m. for 2 h. 100 mg of the dried supernatant were dispersed in 2 mL $H_2O$ and tip sonicated for 2 h at 30 W amplitude, yielding EB-NS.

**Synthesis of LPS-binding-(bLPS)-SWCNTs.** A 13 amino acid long (KKNYSS-SISSIHC) peptide, which was reported to bind lipopolysaccharides (LPS)[42,44], was conjugated to a ssDNA-SWCNT. The LPS-binding peptide was synthesized via solid phase synthesis, precipitated with diethyl ether analyzed by high resolution mass spectrometry. 100 µL (2 mg/mL in 1 x PBS) $(GT)_{20}$-$C_6$-$NH_2$ ssDNA was mixed with 26.8 µL (6 mM in acetonitrile) SMCC (succinimidyl 4-(N-mal-eimidomethyl)cyclohexane-1-carboxylate) (molar ratio of ~ 1:10) and 125 µL 1x PBS (pH 7.4) and left for 1 h reaction at room temperature. Unreacted SMCC was excluded with a 7 kDa desalting column (Zebra™ Spin Desalting Columns, Thermo Scientific). The SMCC coupled ssDNA was directly used for SWCNT surface modification, by adding 75 µL SWCNTs (2 mg/mL in PBS) and sonicating the mixture for 20 min at 25% amplitude, followed by 2 × 30 min centrifugation at 16,100 × g. The non-absorbed ssDNA was removed by sequential molecular cut-off filtration, while concomitant quantifying the bound DNA, following a previously described method by Nißler et al.[60]. The DNA-SWCNT filter-pellet was redispersed in 250 µL PBS by 30 s tip sonication (25% amplitude), followed by 10 min centrifugation (15,000 × g). According to the determined amount of bound SMCC-ssDNA on the SWCNTs, freshly reduced LPS-binding peptides was added with a molar ratio of 1:1 to a SMCC-$(GT)_{20}$-SWCNT solution with 0.8 absorbance at 993 nm and left overnight (12 h) for reaction at room temperature, while continuously shaking. Reduction of the LPS-binding peptide was carried out in PBS, by using an ~1:10 excess of TCEP (Tris(2-carboxyethyl)phosphine hydrochloride). The final LPS-sensor (bLPS-SWCNT) was obtained after 20 min centrifugation (16,100 × g).

**PEG-DA-Hydrogels.** Poly(ethylene glycol) diacrylate (PEG-DA) ($M_n = 700$) was polymerized with 2-hydroxy-4′-(2-hydroxyethoxy)−2-methylpropiophenone (photoinitiator) to create a stable scaffold for the SWCNT nanosensors. To vary the diffusion of the hydrogel (HG), two types of HG-formulas were used (see Supplementary Table ST1 and ST3). Type-I-HG were prepared by mixing 100 mg/mL PEG-DA with 0.5 mg/mL 2-Hydroxy-4′-(2-hydroxyethoxy)−2-methylpropiophe-none in 1x PBS. The photoinitiator was dissolved in $H_2O$ (12 mg/mL) while shaking at 60 °C for 15 min. Macroporous PEG-DA-HGs were obtained by fol-lowing a previously described approach[52]. 100 mg/mL PEG (6 kDa) was added to the hydrogel solution, creating a type-II-HG via polymerization induced phase separation.

After evacuating and purging the liquid HG-solution with $N_2$, the surface-modified SWCNTs were added, characterized with UV-Vis-NIR absorption spectroscopy and directly polymerized in a 1 mL syringe, using an UV-chamber (Belichtungsgerät 1, 4 × 8 W, isel). The SWCNTs-PEG-DA-HG cylinder where dialyzed in 1× PBS for several days to exclude unreacted educts. A typical formulation to yield 5 ml SWCNTs-PEG-DA-HG is given in Supplementary Table ST1.

**Pyoverdine extraction.** *Pseudomonas fluorescens* ATCC 13525 (Supplementary Fig. S3) was cultivated in iron-deficient succinate medium for 4 d at 25 °C/200 rmp[78]. Cultures were centrifuged and sterile-filtrated, before performing solid phase extraction of pyoverdines[79,80]. The supernatant was adjusted to pH 6 with NaOH and passed through (~100 g) Amberlite XAD-4. The resin was washed with 500 mL $H_2O$, and the pyoverdine fraction eluted with 300 ml 80% MeOH: $H_2O$. MeOH was removed from the mixture by evaporation, followed by a liquid-liquid extraction (3 × 50 mL) with $CHCl_3$. Lyophilization yielded the crude extract, which was resuspended in 20 mL of $H_2O$ and applied to an (10 g / 70 mL) washed and pre-conditioned $C_{18}$ec SPE column (Macherey-Nagel GmbH). After a washing step with 50 mL $H_2O$, pyoverdines were fractionally eluted with 10% MeOH in $H_2O$ and lyophilized.

**NIR spectroscopy.** Absorption spectroscopy was performed with a JASCO V-670 device from 400 to 1350 nm in 0.2 nm steps in a 10 mm path cuvette. The NIR fluorescence spectra were acquired with a Shamrock 193i spectrometer (Andor Solis Software (version 4.29.30012.0), Andor Technology Ltd., Belfast, Northern Ireland) connected to a IX53 Microscope (Olympus, Tokyo, Japan). Excitation was performed with a gem 561 laser (Laser Quantum, Stockport, UK), Cobolt Jive laser (Cobolt AB, Solna, Sweden) and Monochromator MSH150, equipped with a LSE341 light source (LOT-Quantum Design GmbH, Darmstadt, Germany).

NIR fluorescence analyte response measurements were performed, unless otherwise stated, by addition of 20 µL analyte to 180 µL 0.2 nM SWCNT solution in PBS. Data analysis was performed with GaphPad Prism 8.3 and OriginPro 9.1.

**SWCNT separation.** Separation of (6,5)-SWCNTs was performed according to a previously reported aqueous two-phase extraction (ATPE) protocol from Li et al.[81]. Briefly, in a three step approach SWCNT chiralities were separated between two aqueous phases, containing dextran (MW 70000 Da, 4% (by mass)) and PEG (MW 6000 Da, 8% (by mass)) with varying pH-values due to HCl addition. The final B3 (bottom)-phase yielded near monochiral (6,5)-SWCNTs, which were diluted with DOC to obtain a stable 1% DOC-SWCNT solution. Further dialysis with a 300 kDa dialysis bag against 1% DOC removed the dextran polymer, used for SWCNT separation. Surface exchange of the (6,5)-SWCNT towards LPS-binding peptide conjugated $(GT)_{20}$ ssDNA was achieved by applying the steps from Streit et al.[82]. Here, 150 µL of purified (6,5)-SWCNTs in 1% DOC (~2 absorption at 986 nm) were used with 25 µL of PEG (MW 6 kDa, 25% ($m/V$)) and 30 µL of conjugated DNA (2.5 mg/mL in $H_2O$). After one precipitation cycle, the nanotube pellet was directly redispersed in 200 µL 1xPBS and characterized by absorption spectroscopy.

HiPco-SWCNTs (Nano Integris, HiPco Raw SWCNTs, 2 mg/mL) were dispersed in 1% DOC for 20 min at 30% amplitude and centrifuged 2 × 30 min at 16,100×g. Using the ATPE protocol[81], large SWCNT diameters were separated from the raw SWCNT mixture. Using a 40 mL formulation with differing SDS concentration (0.375%), for the first separation 43 µL HCl (0.5 M) was added, mixed and centrifuged. Subsequently the T1 (top)-phase was transferred to the B1-mimic and mixed with 8 µL NaOH (0.5 M). Again, the top-phase (T2) was transferred to a fresh B1-mimic and mixed with 20 µL NaOH, which yielded after centrifugation the desired B3-phase with a fraction of large SWCNT chiralities (mainly (9,4), (8,6), and (9,5)-SWCNTs). Dialysis with a 300 kDa dialysis bag against 1% DOC removed the dextran polymer. For further surface exchange to PEG-phospholipids[33] (18:0 PEG5000 PE), the large-chirality SWCNT sample was concentrated in a 100 kDa molecular weight cut-off filter, washed and redispersed in 800 µL sodium cholate (12 mg/mL in PBS). 2.5 mg 18:0 PEG5000 PE was dissolved in 200 µL PBS and mixed with the large-chirality SWCNT fraction, flowed by dialysis against 1 × PBS, using a 1 kDa cut-off bag.

**NIR stand-off imaging of sensor gels.** NIR stand-off detection was performed with a custom made, portable setup, using a XEVA (Xenics, Leuven Belgium) NIR optimized InGaAs camera (Kowa objective, $f = 25$ mm/F1.4) and a white-light source (UHPLCC-01, UHP-LED-white, Prizmatix) equipped with a 700 nm short pass filter (FESH0700, ThorLabs) for excitation. Optical filters (FEL0900, FEL0950, FB1000-10, FB1250-10 ThorLabs) in a manual filter wheel (CFW6/M, ThorLabs) were mounted in front of the camera, which was equipped with an additional 900 nm long pass filter (FEL0900, ThorLabs). Stand-off distance for NIR fluorescence detection for the hydrogel-array experiments (1 s integration time, light intensity 54 mW cm⁻²) was 25 cm and 10 cm for the hyperspectral imaging (5 s integration time, light intensity 18 W cm⁻²). Light intensity was measured at 570 nm with a power meter (PM16-121, ThorLabs).

For evaluation of the sensor responses sensor gels were placed inside a 12-well plate and incubated with the appropriate buffer. Unless otherwise stated 1 × PBS pH 7.4 was used. DNase I (PanReac AppliChem, 5160.7 U/mg) and microbial nuclease (S. aureus, N5386 Sigma Aldrich) was tested in 10 mM Tris-HCl pH 7.5 (2.5 mM $MgCl_2$, 0.1 mM $CaCl_2$), Proteases (S. griseus, P5147 Sigma Aldrich) and Endoproteinase (Glu-C from S. aureus V8, P2922 Sigma Aldrich) was tested in 50 mM Tris-HCl pH 7.5. Thermal inactivation and denaturation of enzymes were performed by heating the desired solution up to 95 °C for 20 min under continuous shaking.

**Assembly of the SWCNT-hydrogel array.** 1.5 cm long hydrogel cylinders of all nine different nanosensors were placed in a cubic (1.5 cm) glass reaction chamber, sealed with parafilm and filled with 1 ml type-I-HG. UV-curing (Belichtungsgerät 1, 4 × 8 W, isel) was performed 8 min for each top and down side. The resulting HG block was sliced into 0.8 mm thin layers, using a specifically designed alumina cutting chamber and razor blades (Supplement Fig. S9). All nanosensor arrays were stored in 1 × PBS to remove non-reacted monomers. HG array sterilization was performed by multiple exchange of sterile buffer and UV-sterilization. Then, the hydrogel arrays in sterile PBS were placed under a sterile hood (TELSTAR AA-30/70) and were illuminated from the top (UV sterile hood, DRI SHIM 30T8/GL) and the bottom (UV-Kontaktlampe Chroma41, 254 nm, Vetter GmbH) with UV light with 2× buffer exchange for 20 min.

For sensor array response analysis during bacterial growth, the sterile hydrogel arrays were fixed with a small amount (~150 µL) 1.5% agarose to the bottom of sterile Petri dishes and overlaid with ~2 mm microbiological agar (total of 5 ml LB-agar with 5% FCS (fetal calf serum, FCS premium, bio west) and 2.5 mg/L Amphotericin B (Biodrom GmbH)), followed by a further UV-sterilization step. HG arrays cast in microbiological agar were stored at 4 °C until usage.

**Image analysis.** NIR images were acquired with Xeneth Software 2.7 (Xenics, Leuven Belgium) and converted in ImageJ (1.51k) into 8-bit data format. The intensities of the HG nanosensors were evaluated with a circular region of interest,

matching the size of the individual HG spot. The mean intensity value of each spot was measured at different timepoints ($I$) and referenced to its start intensity ($I_0$) as $(I-I_0)/I_0$. For HG array experiments, the mean intensity of each nanosensor spot was referenced to the EB-NS intensity, and further comparison of this ratio between different timepoints lead to the sensor response $\Delta I_{SR}$:

$$\Delta I_{SR} = \frac{I_{S1}}{I_{R1}} / \frac{I_{S0}}{I_{R0}}. \tag{1}$$

Here, $I_S$ is the intensity of a specific sensor and $I_R$ the intensity of the EB-NS reference at timepoint ($t=1$) compared to the start ($t=0$). Sensor spots for hyperspectral imaging, were background corrected using an equal size area close to the sensor spots. Principle component analysis (PCA) was performed in R (version 3.6.1) using the package ggbiplot (version 0.55).

**Bacterial strains.** Reference isolates were purchased from the Leibniz Institute DSMZ-German Collection of Microorganisms and Cell Cultures GmbH. Clinical isolates were taken from routine diagnostics of the University Medical Center Göttingen. If available, isolates stemmed from native joint infection, implant loosening, or peri-prosthetic joint infection or related clinical samples. Supplementary Table ST5 summarized the bacteria strains used for pathogen differentiation experiments. Briefly, one reference strain for each species (except *S. pyogenes*) plus one or two fresh clinical isolates were used (*Staphylococcus aureus* ($n=3$), *Pseudomonas aeruginosa* ($n=3$), *Enterococcus faecalis* ($n=3$), *Staphylococcus epidermidis* ($n=2$), *Escherichia coli* ($n=3$), and *Streptococcus pyogenes* ($n=3$)). The strain set, for *S. epidermidis* and *S. aureus* differentiation, was composed of >20 isolates for each species (distinct from strain set I, listed in Supplementary Table ST6). *S. epidermidis* and *S.aureus* isolates were MLST-typed[83,84] specifically for this purpose and the data uploaded to pubmlst.org. When available, *S. aureus* spa-typing/MLST data was taken from previous routine diagnostic procedures. The strain set for hyperspectral bacteria differentiation (three biological and three technical replicates, $n=3$) was composed of *S. aureus* ATCC 29213, isolate PEU3438, isolate PEU3437 and *P. aeruginosa* ATCC 27853, isolate PEU3441, and isolate PEU3440.

**Bacterial detection procedure.** Bacterial collection strains (Supplementary Table ST5 and ST6) were thawed and passaged twice overnight on Columbia blood-agar (Oxoid). Single colonies were picked and diluted with sterile 0.7% NaCl to 0.5 McFarland turbidity. Microbiological agar (LB-agar with 5% FCS and 2.5 mg/L Amphotericin B, plate diameter 5.4 cm) embedded with a single HG array were inoculated with 100 μL bacterial suspension, evenly plated, and incubated at 37 °C. In defined time intervals (1, 2, 4, 8, 12, 24, 36, 48, 72 h), the nanosensor NIR fluorescence was measured using the portable stand-off detection system with 1 s integration time. For each strain, three stated biological replicates within three technical replicates each were tested.

Liquid cultures of *S. aureus* ATCC 29213 and *P. aeruginosa* isolate PEU3440 were obtained by inoculating 25 ml LB-media with a single colony from a fresh overnight-culture (Columbia blood-agar). After 24 h incubation at 37 °C and constant shaking, ($OD_{600}$ *S. aureus* 2.94; $OD_{600}$ *P. aeruginosa* 0.86) 2 × 20 min centrifugation and further sterile filtration (0.45 μm) yielded a cell-free supernatant. For each condition, a sensor array was conditioned by 1 h incubation in sterile LB-medium in a 5.4 cm sterile petri dish, the medium then replaced by 5 ml culture supernatant, and NIR fluorescence images acquired in 30 s intervals.

**Human joint fluids (synovia).** Synovial liquid samples from human knee joints were collected after written consent was obtained from all patients (ethic proposal number 311/18, approved by medical faculty's ethic committee, University of Bonn). Samples were taken intraoperatively during surgery due to native joint infection, implant loosening, or peri-prosthetic joint infection as part of standard diagnostics for microbiologic and pathologic analysis. A small portion from each sample was kept for scientific analysis and samples were shock-frosted and stored at −80 °C.

350 μL human synovia was directly applied to the sensor arrays, which were incubated 1 h beforehand in 0.9% NaCl solution. 13 independent samples from high-grade infections, five independent samples of low-grade infections and 8 samples from patients without diagnosed infections (infection classification based on the clinical report) were analyzed. pH of the synovia was tested by adding 20 μL to a pH-indicator paper (Dip in, pH 0–14, VWR). Sensing of bacterial targets with varying synovia background was performed by using three independent samples for non- and high-grade infections and analyzing the sensor response towards pH 4.5 and protease activity (from *S. griseus*, 100 μg/mL).

**Stochastic simulation of sensor responses for bacteria differentiation and classification.** For a number of bacteria species n_B a response pattern for the n_S sensors is randomly generated in a given range of responses modeled after values from the measurements. In this simulation up to n_B = 10 bacteria and n_S = 25 sensors were initiated. The response of the sensor set is either a uniformly positive or negative response (sensor responses = 0.7–1.7). A given number of sensors per set produce the same response pattern as they do for another bacterium, therefore are set to the same random value. According to the measurement

data, approximately 40% of the sensors had an indistinguishable response compared the dataset of another bacterium. The response of the rest of the sensors was randomly chosen within the known experimental range for the different bacteria. To the response matrix a random noise is applied $r$ times to account for experiment repetition. The noise observed in the data was up to 10%. For the principal component analysis (PCA), the number of sensors is equal to the number of principal components (PC), therefore the experiment was virtually repeated 25 times. When considering a rising number of PC, the response matrix is generated in its entirety and a rising number of senor entries are used in the calculation to model the development of additional sensors. The number of features in a PCA must be equal or exceed the number of PC, therefore $r$ is equal to n_S to generate a doubled dataset for training and testing the PCA. The PCA as implemented in scikit-learn[85] is solved with a full singular value decomposition (SVD) and a logistic regression with a bilinear solver and is used to predict the bacteria species of the test dataset. The percentage of correctly assigned test cases can be calculated with a confusion matrix which has correct assignments as its diagonal elements. Used Python packages: matplotlib[86] (version 3.0.3), NumPy[87] (version 1.16.2), scikit-learn[85] (version 0.20.3), Pandas (version 0.24.2).[88]

**Reporting summary.** Further information on research design is available in the Nature Research Reporting Summary linked to this article.

## Data availability
The main data supporting the results of this study are available within the paper and its Supplementary Information. The related source data files are available under https://doi.org/10.5281/zenodo.4072999. Data on bacterial strains were made accessible online (https://pubmlst.org/bigsdb?db=pubmlst_sepidermidis_isolates&page=query; https://pubmlst.org/bigsdb?db=pubmlst_saureus_isolates&page=query) as indicated in the manuscript. Information on the bacterial strain identification is available in Supplementary Table 6.

## Code availability
The Python code for stochastic sensor simulation is described in the manuscript and is available on https://gitlab.gwdg.de/m.dohmen/bacteria-sensing.git.

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

## Acknowledgements

This project was supported by the VW foundation. We furthermore acknowledge support by the German Research Foundation within the Cluster of Excellence RESOLV EXC2033 and the Heisenberg program (S.K.). We thank Prof. Dr. Andreas Janshoff and Prof. Dr. Claudia Steinem and their groups for fruitful discussions and support. We thank Hans-Joachim Heymel for the construction of the hydrogel cutting chamber, Dr. Elena Polo for peptide synthesis, Agnieszka Goretzki, Yvonne Laukat, and Nina Gerkens for help with bacteria cultivation and *MLST* typing, Dr. Thomas Randau, Dr. Sascha Gravius, and Dr. Frank A. Schildberg for intraoperative synovia collection, Seren Hamsici for help with initial LPS sensor development, Larissa Kurth for SWCNT purification and Dr. Ingo Mey for help during SEM measurements.

## Author contributions

R.N. and S.K. designed and conceived the research S.K. coordinated the project. O.B., C.N., and U.G. collected the clinical bacteria isolates, characterized them, and helped with bacterial detection experiments. S.G.W. helped with human synovia collection and related sensing experiments. G.S. exfoliated EB-NS and performed SEM characterization. M.D. developed the simulation with input from S.K. All other experiments were performed by R.N. All authors contributed to the writing of the manuscript and analysis of data.

## Funding

## Competing interests

The authors declare no competing interests.
