## [Peer Review File · Nature Communications]

Reviewers' Comments:

Reviewer #1:

Remarks to the Author:

The manuscript describes in detail and in depth the design and implementation of a set of near infrared (NIR) fluorescent nanosensors to be used for remote fingerprinting of clinically important bacteria. The work is very rigorous, and the results are interesting and convincing regarding the presence and the identification, by the detection of specific metabolites, of different pathogens. While the bacterial detection in hydrogels is accurate and very satisfactory, the possibility of bacterial detection and identification in situ is limited by the short penetration of the fluorescent nanosensors into the tissues. To demonstrate the applicability of the described technology in the case of implant-associated infections, the Authors tested the sensor array performance in the presence of human synovial fluid, obtaining satisfactory results. The Authors conclude that "the sensor array could respond towards a local, biofilm-based infection, while background signals in synovia would not lead to a false-positive readout". The authors conveniently warn that, for a future implant application and in situ diagnostics, the advantage or the limit of NIR is the penetration into tissues, which they estimate to be about 7 mm. For these considerations, the expression "in situ" appears excessively optimistic in the title of the paper, which, at the state of the art of this methodology, is rather a target than a result. Overall, the manuscript deserves to be published with a minor revision, since the Authors should use more caution in affirming the possibility of remote in situ detection of pathogens in consideration of the limited tissue penetration of sensors.

Reviewer #2:

Remarks to the Author:

This is a very nice paper in which single-walled carbon nanotube (SWCNT)-based NIR nanosensors have been developed for remote fingerprinting of clinically important bacteria through detection of virulence factors and metabolites that they exude. Non-covalently modified SWCNTs are chosen that detect released metabolites. These are integrated into functional hydrogel arrays in a few different ways (in solution, spatially in hydrogels, or by hyperspectral imaging). One of the advantages is that detection is much faster than existing techniques such as those based on DNA/PCR, Mass Spectroscopy, even with microfluidic enhancement. The system was tested on six important bacteria.

1. Part of the innovation here is that sensing is remote using an NIR camera. The authors suggest that they could detect post-operative infections inside the body as in artificial joint implants.

2. The problem they have chosen – bacteria detection – poses a difficult sensing problem because of the highly complex biochemical processes, and access to the site where they need to be measured.

3. One sensor is not selective enough and the concept of a chemical nose appears to be more promising. Having said that, their data analysis is relatively simple. There are now many much more sophisticated machine learning algorithms available that the authors could employ. Some discussion of this point should be included.

4. They used eight sensors, four tailored for specific bacterial targets and four generic lower-selectivity sensors. This combination of data-driven and "rational" design is a good one.

5. For long-term stability and robustness, the authors incorporate the nanosensors into PEG hydrogels with different porosity. Tested on protease digestion of BSA on SWCNT (decreases intensity) and nucleases.

6. The authors demonstrate spatial encoding of sensors. They also make use of the non-interacting nature of SWCNT fluorescence to develop a Hyperspectral NIR procedure.

Overall, I found the paper to be very well written and the work complete. It is a significant advance for the field and I am happy to recommend it for publication.

Reviewer #3:

Remarks to the Author:

In the article 'Remote near infrared in situ identification of pathogens with multiplexed nanosensors' a hydrogel-based sensor-platform containing single walled carbon nanotubes (SWCNT) as reporters for the identification of six different pathogens is presented. The sensor platform consists of a hydrogel array of nine different 'hydrogel spots' loaded with SWCNT with different recognition coating, four of them reflect specific recognition and four unspecific recognition, and identification of the pathogens is conducted by a specific fluorescence response pattern of the hydrogel array. In addition to the spatial patterns, identification of pathogens was also shown via hyperspectral imaging.

The combination of specific and unspecific sensors to form specific response patterns is interesting and also the proof of concept for hyperspectral imaging and in-tissue measurements looks promising.

The work is overall interesting and is well fitted to Nat. Comm., therefore I recommend publication given the authors address the following issues:

- The main argument of the article, that pathogen recognition can be performed by forming a visible response pattern via intensity changes of the SWCNTs on the hydrogel array after an incubation time of 72 h is shown nicely in fig 4d and 4f, (and maybe a bar-plot presentation of figure S12, data points after 72h) and is convincing. However, some of the panels are too crowded and the data points and lines cannot be distinguished (4 e and g) or that do not seem to be necessary (figure 4h time series, after 72h is enough and figure 4k and l).
- The advantage of principle component analysis at this stage is not clear, as it also does not manage to distinguish between *E. faecalis* and *S. pyogenes*.
- The article starts by introducing two of the specific sensors, then explains the hydrogel-matrix and then goes back to present two additional specific sensors, which is confusing and would profit from some restructuring.
- The LPS sensor in solution is highly responsive on *E. coli* in and less responsive on *P. aeruginosa* while figure 4f indicates that this is reversed in the hydrogel. In general, it would be nice to discuss the response of the sensor towards the different pathogens. Is there an educated guess on how they should react? Are the siderophores tested on HeApta abundant in the selected bacteria?
- Table ST3 should be added to the main text.
- The sensor response of the specific sensors can be shown in a more unified way. Figure 3 can be displayed in line plots as well, as the different time intervals do not add information (except of Fig 3 c).
- Adding the unspecific sensors is a nice idea, and its benefit should be emphasized and would strengthen the article.
- The figures need revisions in terms of visibility.

Detailed response to reviewer comments

Reviewer comments are shown in black and our response in blue. In the revised manuscript changes are highlighted in yellow.

Reviewer #1 (Remarks to the Author):

“The manuscript describes in detail and in depth the design and implementation of a set of near infrared (NIR) fluorescent nanosensors to be used for remote fingerprinting of clinically important bacteria. The work is very rigorous, and the results are interesting and convincing regarding the presence and the identification, by the detection of specific metabolites, of different pathogens. While the bacterial detection in hydrogels is accurate and very satisfactory, the possibility of bacterial detection and identification in situ is limited by the short penetration of the fluorescent nanosensors into the tissues. To demonstrate the applicability of the described technology in the case of implant-associated infections, the Authors tested the sensor array performance in the presence of human synovial fluid, obtaining satisfactory results. The Authors conclude that “the sensor array could respond towards a local, biofilm-based infection, while background signals in synovia would not lead to a false-positive readout”. The authors conveniently warn that, for a future implant application and in situ diagnostics, the advantage or the limit of NIR is the penetration into tissues, which they estimate to be about 7 mm. For these considerations, the expression "in situ" appears excessively optimistic in the title of the paper, which, at the state of the art of this methodology, is rather a target than a result. Overall, the manuscript deserves to be published with a minor revision, since the Authors should use more caution in affirming the possibility of remote in situ detection of pathogens in consideration of the limited tissue penetration of sensors.”

Answer: We thank the reviewer for the positive feedback. In general, we agree that in vivo applications are challenging but at the same time extremely relevant. Therefore, we rephrased respective parts in the manuscript, highlighting the remaining challenges regarding stand-off detection and tissue penetration. Our manuscript shows the huge potential of a novel approach and we strongly believe that improved optics such as using higher excitation energy (e.g. a laser instead of a LED) or excitation/detection schemes (pulsed excitation, life-time imaging,...) will further increase tissue penetration, sensitivity etc. Consequently, we are convinced that the novel insights and advances in this work should be used to coin the term ‘in situ’ to summarize the main ideas/concepts and attract more researchers to work in this important field in the future.

Reviewer #2 (Remarks to the Author):

“This is a very nice paper in which single-walled carbon nanotube (SWCNT)-based NIR nanosensors have been developed for remote fingerprinting of clinically important bacteria through detection of virulence factors and metabolites that they exude...

...3. One sensor is not selective enough and the concept of a chemical nose appears to be more promising. Having said that, their data analysis is relatively simple. There are now many much more sophisticated machine learning algorithms available that the authors could employ. Some discussion of this point should be included.

Answer: We thank the reviewer for the enthusiasm and positive comments on our work. We agree that there are indeed sophisticated machine learning algorithms. They could enable even faster and more precise classification and prediction of bacterial contamination in the future. Nevertheless, even the relatively simple multivariate analysis (PCA) allowed us to distinguish bacteria based on their chemical NIR fingerprint, which highlights the high sensitivity of the approach. We expanded the discussion of advanced algorithms to analyze sensor data sets in the revised manuscript to account for this important hint of the reviewer.

Reviewer #3 (Remarks to the Author):

1) “The work is overall interesting and is well fitted to Nat. Comm., therefor I recommend publication given the authors address the following issues...”

Answer: We thank the reviewer for the encouraging comments and hints.

2) “The main argument of the article, that pathogen recognition can be performed by forming a visible response pattern via intensity changes of the SWCNTs on the hydrogel array after an incubation time of 72 h is shown nicely in fig 4d and 4f, (and maybe a bar-plot presentation of figure S12, data points after 72h) and is convincing. However, some of the panels are too crowded and the data points and lines cannot be distinguished (4 e and g) or that do not seem to be necessary (figure 4h time series, after 72h is enough and figure 4k and l).”

Answer: We thank for this comment and agree that there is a lot of data presented in figure 4. Figure 4 e and g are supposed to illustrate the complexity but already highlight the differences between the bacterial sensor responses. The complexity of the data then motivates the use of principal component analysis in figure 4h, which leads to a graphical representation of bacteria clusters. In the revised manuscript we rearranged some of the sub-figures and enlarged the captions to improve the readability.

3) ”The advantage of principle component analysis at this stage is not clear, as it also does not manage to distinguish between *E. faecalis* and *S. pyogenes*.”

Answer: PCA was used for multivariate statistics to analyze the bacterial fingerprints and figure 4 clearly shows that it is possible to distinguish important bacteria. As explained above it allows a much better cluster analysis than the raw data plots in figure 4 d,f. The reason why *E. faecalis* and *S. pyogenes* cannot be distinguished is not because of the statistical analysis but rather the very similar chemical fingerprints (see SI Figure S12). In the future, additional sensors focusing on the differences between these two bacteria could make this discrimination possible.

4) ”The article starts by introducing two of the specific sensors, then explains the hydrogel-matrix and then goes back to present two additional specific sensors, which is confusing and would profit from some restructuring.”

Answer: We thank the reviewer for pointing us to this issue. The reason for this order is the chemical design. The two (additional) specific sensors for proteases and nucleases only work within a hydrogel because in solution degradation of the functionalization would cause aggregation and precipitation of the SWCNTs. In contrast, the rationally designed LPS- and siderophore sensors work in solution and hydrogel. We clarified this aspect in the revised version.

5) “The LPS sensor in solution is highly responsive on *E. coli* in and less responsive on *P. aeruginosa* while figure 4f indicates that this is reversed in the hydrogel. In general, it would be nice to discuss the response of the sensor towards the different pathogens. Is there an educated guess on how they should react? Are the siderophores tested on HeApta abundant in the selected bacteria?”

Answer: The LPS sensor was tested against the direct chemical targets i.e. lipopolysaccharides of gram-negative bacteria and then directly in the complex experiment with medium, bacteria and multiple sensors. Therefore, the absolute fluorescence response against isolated LPS is not directly comparable to the complex environment of the realistic detection experiment. The siderophore target is indeed a main virulence factor in the most of the tested bacteria (*E. coli*, *S. aureus*, *P. aeruginosa*...).¹ Especially the tested pyoverdines are structurally closely related to the ones known from *P. aeruginosa*.²

6) “Table ST3 should be added to the main text.”

Answer: We think that ST3 would be misleading in the main manuscript because it could convey the message that the sensors would work exactly with bacteria as they do *in vitro* with pure target molecules. The table is mainly a summary of the design ideas without bacteria. The main point of the paper is that by combining many sensors including ‘non-selective’ ones the overall sensitivity is enhanced. Therefore, we prefer to keep this table in the supplementary information.

7) ”The sensor response of the specific sensors can be shown in a more unified way. Figure 3 can be displayed in line plots as well, as the different time intervals do not add information (except of Fig 3 c). “

Answer: Figure 3 shows the hydrogel-specific data in a unified way, including presented as bar plots at different timepoints. Even if the fluorescence changes between different timepoints are absent (in two sensors), it conveys the message for the reader, that different sensor types react not only to various analytes, but also on different timescales. Consequently, we believe that this information is useful for the reader.

8) “Adding the unspecific sensors is a nice idea, and its benefit should be emphasized and would strengthen the article.”

Answer: We thank the reviewer for this suggestion and highlighted this aspect in several parts of the manuscript.

9) ”The figures need revisions in terms of visibility.”

Answer: We improved the clarity and readability in the revised manuscript (e.g. figure 5 a and 5 b to clarify the three different readout spectral channels, improved legends in figure 3.,...).

References

- (1) Holden, V. I.; Bachman, M. A. Diverging Roles of Bacterial Siderophores during Infection. *Metallomics* **2015**, 7 (6), 986–995.
- (2) Budzikiewicz, H. Secondary Metabolites from Fluorescent Pseudomonads. *FEMS Microbiol. Lett.* **1993**, 104 (3–4), 209–228.